# The Safety and Efficacy of Hepatic Transarterial Embolization Using Microspheres and Microcoils in Patients with Symptomatic Polycystic Liver Disease

**DOI:** 10.3390/jpm12101624

**Published:** 2022-10-01

**Authors:** Alexis Coussy, Eva Jambon, Yann Le Bras, Christian Combe, Laurence Chiche, Nicolas Grenier, Clément Marcelin

**Affiliations:** 1Department of Radiology, Pellegrin Hospital, Place Amélie Raba Léon, 33076 Bordeaux, France; 2Departement of Nephrology, Pellegrin Hospital, Place Amélie Raba Léon, 33076 Bordeaux, France; 3Department of Digestive surgery, Haut Leveque, 33076 Bordeaux, France

**Keywords:** embolization, polycystic liver disease, safety, efficacy

## Abstract

**Purpose:** We investigated the long-term safety and efficacy of hepatic transarterial embolization (TAE) in patients with symptomatic polycystic liver disease (PLD). **Materials and Methods:** A total of 26 patients were included, mean age of 52.3 years (range: 33–78 years), undergoing 32 TAE procedures between January 2012 and December 2019 were included in this retrospective study. Distal embolization of the segmental hepatic artery was performed with 300–500 µm embolic microspheres associated with proximal embolization using microcoils. The primary endpoint was clinical efficacy, defined by an improvement in health-related quality of life using a modified Short Form-36 Health Survey and improvement in symptoms (digestive or respiratory symptoms and chronic abdominal pain), without invasive therapy during the follow-up period. Secondary endpoints were a decrease in total liver volume and treated liver volume and complications. **Results:** Hepatic embolization was performed successfully in 30 of 32 procedures with no major adverse events. Clinical efficacy was 73% (19/26). The mean reduction in hepatic volume was −12.6% at 3 months and −27.8% at the last follow-up 51 ± 15.2 months after TAE (range: 30–81 months; both *p*s < 0.01). The mean visual analog scale pain score was 5.4 ± 2.8 before TAE and decreased to 2.7 ± 1.9 after treatment. Three patients had minor adverse events, and one patient had an adverse event of moderate severity. **Conclusion:** Hepatic embolization using microspheres and microcoils is a safe and effective treatment for PLD that improves symptoms and reduces the volume of hepatic cysts.

## 1. Introduction

Polycystic liver disease (PLD) is a group of genetic disorders that manifest as the progressive development of multiple cysts in the liver parenchyma [1,2]. Autosomal dominant polycystic kidney disease (ADPKD) is the most frequent cause of PLD (80%), liver cysts being the most common extrarenal manifestation [3]. Autosomal dominant polycystic liver disease (ADPLD) is a separate entity with two different mutations, responsible for 20% of PLD, with a cystic disease restricted to the liver [4]. Molecular genetic testing is available to look for mutations in the SEC63, LRP5, PRKCS, GANAB, ALG8, SEC61B PKD1, PKD2, and PKHD1.

Gigot classification is now commonly used to define severity in PLD [5]. Type I is defined by the presence of less than 10 large hepatic cysts measuring more than 10 cm in maximum diameter. Type II is defined by a diffuse involvement of liver parenchyma by multiple cysts with remaining large areas of non-cystic liver parenchyma. Type III is defined by presence of diffuse involvement of liver parenchyma by small- and medium-sized liver cysts with only a few areas of normal liver parenchyma. Cyst puncture, sclerotherapy, or fenestration are used to treat with success for Gigot type I. However, Gigot type II and III are more difficult to treat.

Conversely to ADPLD, ADPKD causes progressive renal dysfunction, whereas liver function remains normal in both diseases. However, up to 20% of patients with PLD may require treatment because the compressive effects of cysts on adjacent structures cause progressive symptoms [6]. Common symptoms include abdominal discomfort, such as early satiety and postprandial fullness, chronic and acute pain, dyspnea, reduced mobility, and fatigue [7]. Severe disease can lead to malnutrition and disability.

Different modalities have been reported for treating PLD, with the objective of reducing the liver volume and relieving symptoms [8]. Treatment with somatostatin analog appears to be insufficient, with a decrease in liver volume of only 1.99% after 2 years [9].

Percutaneous cyst aspiration associated with sclerotherapy and laparoscopic fenestration are indicated in patients with superficial cysts and a limited number of large cysts (Gigot classification type I). These treatments may temporarily relieve symptoms, but there is a high recurrence rate of up to 80% [10].

Hepatic resection has been proposed to treat highly symptomatic patients, with diffuse involvement of the liver parenchyma by multiple cysts and remaining large areas of non-cystic liver parenchyma [8]. However, significant complications can occur, and morbidity and mortality rates associated with this procedure can reach 50% and 3%, respectively [11,12].

Hepatic transplantation has been published in cases of diffuse involvement of the liver parenchyma by small and medium cysts with only a few areas of normal parenchyma with good efficacy but a morbidity rate of 40–50% and a global mortality rate between 8% and 17% at 5 years [12].

Transarterial embolization (TAE) of hepatic arteries in PLD was first described as a promising minimally invasive treatment for patients with abdominal discomfort due to a distended abdomen or gastric compression [13]. Several embolization techniques have been described, including coiling and the use of a mixture of *N*-butyl cyanoacrylate–iodized oil [14] and polyvinyl alcohol [15], particles tris-acryl gelatin microspheres [16]. The association between distal and proximal embolization was described in renal embolization in cases of ADPKD, with good efficacy [17]. To date no study evaluated quality of life after the TAE.

The present retrospective study was performed to assess the safety and clinical efficacy of TAE using combined embolization with Tris-acryl gelatin microspheres and coils and its long-term impact on reducing liver volume in patients with symptomatic PLD.

## 2. Patients and Methods

### 2.1. Patient Population 

This retrospective study was approved by the institutional review board, and the requirement for informed consent was waived. A total of 21 patients had PLD and ADPKD (80.7%), whereas 6 had PLD alone. Indication of treatment was based on the following criteria:

Clinically palpable liver hypertrophy responsible for symptomatic mass effects, such as abdominal pain, dyspnea, early satiety, and physical disability

Diffuse disease (Gigot classification type II or III) on imaging precluding surgical resection or percutaneous sclerotherapy

Exclusion criteria included liver cyst infection and the decision of hepatic transplantation taken before TAE.

The decision to perform TAE was made at a multidisciplinary committee based on the patient’s degree of clinical discomfort and on liver imaging data on computed tomography (CT).

All patients were followed until June 2020. Quality of life was determined with a modified Short Form-36 Health Survey (SF-36) by mail or e-mail in June 2020 [18,19]. The SF 36 questionnaire is a health survey frequently used on clinical studies to assess health related quality of life. It included 36 questions in 8 domain scores of physical and mental function. 

Quality of life and the efficacy of TAE for treating digestive and respiratory symptoms and chronic pain were evaluated retrospectively before and after TAE.

Patients’ demographic data, including weight (after dialysis in patients receiving dialysis) and laboratory data were obtained from their electronic medical records. All data were collected before TAE, 3 months after treatment, 2 years, between 2–5 years, and at the last follow-up more than 5 years in June 2020. All medical and surgical treatments undergone by patients before and after TAE were noted.

### 2.2. Imaging before the Procedure

Pre- and post-TAE total liver volume was calculated on CT with or without contrast injection with validated open source image processing software (Osirix, Pixmeo Sarl, Geneva, Switzerland) [20,21] from the set of contiguous images by the product of liver area, traced manually on each CT with a slice thickness of 5 mm.

### 2.3. The TAE Procedure 

All procedures were performed on two different angiographic units (Allura Xper FD20, Philips, Best, The Netherlands; and Artis Pheno, Siemens Healthcare, Forchheim, Germany) under sedation or general anesthesia. After percutaneous introduction of a 5-Fr sheath (Radifocus Introducer II, Terumo, Tokyo, Japan) into the right or left femoral artery under ultrasound guidance, the celiac trunk and superior mesenteric artery were catheterized with a 4-F artery catheter (SHK 1.0, Cordis, Miami, FL, USA; or Cobra C2 Glidecath, Terumo) and a 0.035″ hydrophilic guidewire (Terumo). After contrast injection in each trunk, digital subtraction angiography allowed visualization of the arterial anatomy, and portal hepatography was used to determine the anatomy and permeability of the intra- and extrahepatic portal system (Figure 1). Superselective catheterization of arterial branches vascularizing hepatic segments containing liver cysts was achieved with a 2.7-Fr microcatheter (Progreat, Terumo).

The hepatic regions for embolization were selected before the TAE procedure based on the distribution of cyst density determined on CT. According to these criteria, total or partial liver embolization was performed. Arterial occlusion of each segmental artery was performed first by distal embolization with 300–500 µm Tris-acryl gelatin microspheres (Embosphere, Merit Medical, South Jordan, UT, USA; or Embogold, Boston Scientific, Natick, MA, USA). Particles were injected through the microcatheter until stasis of feeding arterial flow without reflux. Subsequently, to ensure complete vascular occlusion and prevent revascularization, embolization was completed with one or several microcoils of suitable diameter and length (Tornado or Hilal, Cook Medical, Indiana, IN, USA). Diameter of coils were between 2 and 6 mm. When necessary, extrahepatic collaterals, such as the inferior phrenic or omental arteries, were also embolized. 

### 2.4. Postprocedure Management

To prevent postembolization syndrome (PES), a combination of corticosteroid (methylprednisolone 1 mg/kg) and an analgesic (acetaminophen 1 g) were injected intravenously 2 h before embolization. PES is characterized by moderate to severe epigastric pain, fever, severe nausea, and vomiting that appear early after embolization. Biologically, there is a biological inflammatory syndrome associated with elevated transaminases and bilirubin. All these abnormalities are transient and spontaneously resolved in a few days. During embolization, pain was managed with intravenous nonsteroidal anti-inflammatory drugs and morphinic analgesic titration if necessary. After embolization, patients were admitted to the nephrology department for 2 to 3 days, and analgesic and corticoid treatment was maintained intravenously for at least 24 h. If additional analgesics were needed, morphine was added and administered through a patient-controlled analgesia pump with antiemetic treatment if necessary.

Laboratory data included Creat, Urea, AST, ALT, GGT, Bilirubin, and Alcaline phosphatase were examined on days 1 and 3 after TAE. After discharge, pain was controlled with acetaminophen or tramadol, and prednisolone (20 mg/day) was given for 8 days.

### 2.5. Endpoints

The primary study endpoint was clinical success, defined as improvement in quality of life (an increase in SF-36 score > 10 points) [22,23] and improvement in abdominal pain, digestive and respiratory symptoms without invasive treatment throughout the follow-up period. The SF-36 was completed in consultation on the follow-up scan was performed. Secondary outcomes were primary technical success, defined by complete occlusion of targeted segmental hepatic arteries; a decrease in liver volume on CT with or without contrast injection and complications according to the guidelines of the Society of Interventional Radiology [24] and Clavien–Dindo [25], including cyst complications, such as hemorrhage or infection during follow-up.

### 2.6. Statistical Analysis

Data are summarized as proportions and means ± standard deviations as appropriate. Categorical variables were analyzed with the chi-square test, and continuous variables were compared with Student’s *t* test or analysis of variance. In all analyses, *p* < 0.05 was taken to indicate statistical significance. If patients could not visit the hospital at the time of data collection, their data were imputed by linear regression based on the next visit.

## 3. Results

### 3.1. Population

From 1 January 2012, to December 2019, a total of 26 consecutive patients (21 women [81%] and 5 men [19%]) with symptomatic PLD underwent a total of 32 TAE procedures at a single institution. Before TAE, 7 patients received treatment with somatostatin for 12–24 months, which was considered ineffective; 11 patients had been treated by cyst puncture and sclerotherapy, but no patients had undergone any previous surgical interventions. Patient characteristics are summarized in Table 1 and Table 2. The mean follow-up period was 51 months (range: 6–98 months). In this population, 16 and 10 patients had Gigot liver type II and type III, respectively. Embolization was performed selectively in 23 procedures (72%)—14 in the right liver (44%) and 9 in the left liver (28%)—and was global (right and left) in 9 procedures (28%). The mean number of embolized segments was 4.0 ± 1.7. Extra-hepatic arteries embolized were three left gastric arteries.

### 3.2. Technical Success

Primary technical success was achieved in 30 of the 32 procedures (93%). TAE was stopped because of difficulty catheterizing segmental branches responsible for a high radiation dose (4 Gy) in one case and dissection of the left liver artery in another case. These two patients underwent a second TAE with effective technical success. Two procedures were necessary in five patients and one patient underwent three procedures because of insufficient reduction in volume. These 6 failures were due to recanalization of hepatic arteries or embolization of insufficient hepatic volume but after the subsequent procedures the secondary technical success rate was 100%.

The mean procedure time was 90.9 ± 28.8 min. The mean quantity of contrast injected during the procedure was 134.5 ± 52 cc, the mean radiation dose area product (DAP) was 1146 ± 935 mGy, and the mean fluoroscopy time was 34.5 ± 14 min.

### 3.3. Safety

TAE postembolization syndrome occurred after all procedures despite medical preparation, but no patients developed hepatic insufficiency. The mean duration of hospital stay was 4.25 ± 1.9 days (range: 2–10 days). No residual pain was reported after 1 month.

The total complication rate was 12.5%. Three patients presented pain recurrence (grade I) justifying new hospitalization between 1 and 10 days. One patient had a cyst infection (grade III-a) that occurred 20 days after TAE, treated successfully with puncture and intravenous antibiotics over a 5-day hospitalization period.

### 3.4. Reduction in Hepatic Volume (Figure 2 and Table 3)

The mean decrease in total liver volume at 3 months was −12.6% (±8.01%) compared to the pre-TAE value or a mean loss of −855 cc (*p* < 0.01). CT was performed more than 2 years after TAE in 12 patients, and the mean decrease in total liver volume ratio was −27.8% (*p* < 0.01) (Table 4).

**Figure 2 jpm-12-01624-f002:**
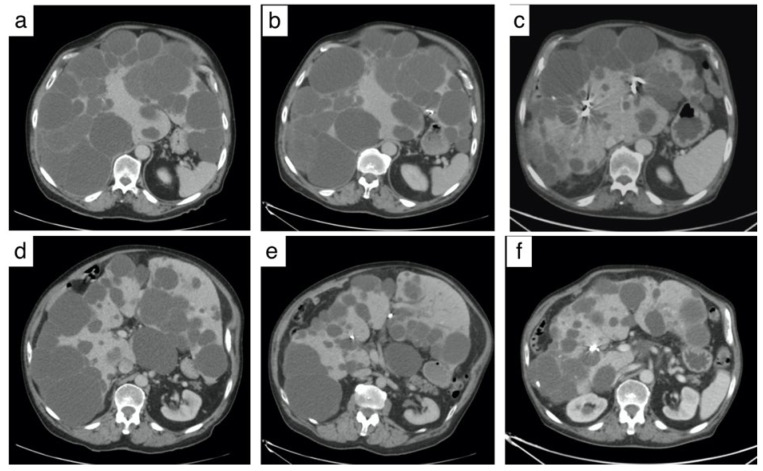
Computed Tomography (CT) from a patient with polycystic liver disease before and after transarterial embolization. (**a**,**b**) Pre-embolization CT showing a voluminous polycystic liver with a compression of the stomach by the cysts. (**c**,**d**) CT at 3 months showing coils and reduction of cysts volume. (**e**,**f**): CT at 4 years showing a significant liver volume reduction of 29% with a decompression of the stomach.

**Table 3 jpm-12-01624-t003:** Liver volume reduction in sub group analysis.

	Sex	Liver Volume before TAE (mL)	Liver Volume after TAE (mL)	Difference		*p*
Total liver	men	8808.4	7441.1	−15.2%	*p* < 0.01	
	women	5672.4	4932.7	−11.9%	*p* < 0.01	
				Qobs = 0.90 IC95% [−4.1169; 10.5965]		*p* = 0.37
Embolized liver	men	5894	4752	−17.8%	*p* < 0.01	
	women	3906	3234.7	−24.3%	*p* < 0.01	
				Qobs = −1.16 IC95% [−17.4903; 4.7764]		*p* = 0.25
	**Disease**					
Total liver	ADPLD	8633	7174	−16.8%	*p* < 0.01	
	ADPKD	5597	4916	−11.3%	*p* < 0.01	
				Qobs = 1.64 IC95 [−1.3294; 12.3023]		*p* = 0.11
Embolized liver	ADPLD	5787	4398	−24%	*p* < 0.01	
	ADPKD	3781	3291	−22%	*p* < 0.01	
				Qobs = 0.39 IC95 [−8.6315; 12.6986]		*p* = 0.69
	**Quality of life**					
Total liver	Improved	6313	5322	−15.7%	*p* < 0.01	
	Not improved	6420	5878	−8.1%	*p* = 0.03	
				Qobs = −2.65 IC95 [−11.7682; −1.5267]		*p* < 0.01
Embolized liver	Improved	4708 cc	3811 cc	−24%	*p* < 0.01	
	Not improved	3543 cc	2857 cc	−12%	*p* < 0.01	
				Qosb = −3.12 IC95 [−11.7682; −1.5267]		*p* < 0.01

**Table 4 jpm-12-01624-t004:** Liver volume reduction at 3 months and 2 years after TAE.

	Mean Vol. before TAE (mL)	Mean Vol. after TAE (mL)	Mean Reduction of Volume	*p*
*Scanner 3 months after TAE*				
Total liver Volume	6438 (+/−2592)	5567 cc (+/−2122)	−12.6% (+/−8) −855 cc IC95% [570.88; 1140.55]	*p* < 0.01
Liver embolized	4623 (+/−2916)	3692 cc (+/−2460)	−22.7% (+/−12.5) −930 cc IC95% [679.03; 1182.78]	*p* < 0.01
*2+ years of follow-up*				
Total Liver Volume (*n* = 12 patients)	6275 (+/−2353)	4440 cc (+/−1302)	−27.8% −1863 cc IC95% [757.7333; 2968.4889]	*p* < 0.01
Liver embolized (*n* = 12 patients)	5001(+/−2157)	3546 cc (+/−1769)	−32.5% (+/−17) −1544 cc IC95 [841.9154; 2246.0846]	*p* < 0.01

### 3.5. Clinical Efficacy

The primary clinical efficacy was 72% (19/26) at the date of evaluation in June 2020, and the average follow-up time was 51 ± 15.2 months (range: 6–98 months). A total of 19 patients (72%) showed a significant improvement in their quality of life with an increase in SF-36 score of >10 points, and 13 (52%) showed an increase of >20 points. No patients reported a worsening of quality of life after embolization. Of the 26 patients, 23 (88%) would recommend TAE for the management of PLD. 

Clinical failure occurred in seven patients (27%) who experienced no improvement in quality of life after TAE. Two of these patients (7%) required complementary surgical treatment 7 and 11 months after TAE consisting of a right hepatectomy in one case and a combination of a right hepatectomy and cyst fenestration in the other. Surgery was effective in these two patients. Five patients did not benefit for another treatment. One patient died of metastatic kidney cancer 2 years after treatment unrelated to TAE. The four other patients who did not experience clinical success underwent repeated cyst aspirations or symptomatic treatment and refused a second TAE or surgery.

#### 3.5.1. Chronic Pain

Of the 24 patients (92%) who presented with chronic pain related to liver volume, 20 (83%) noticed an improvement after TAE. Pain evolved from a mean visual analog scale score of 5.4 ± 2.8 before TAE to 2.7 ± 1.9 afterward.

#### 3.5.2. Digestive Symptoms

All patients had digestive symptoms before TAE (moderate in 17 and severe in 9). After TAE, 21/26 patients (81%) showed an improvement in symptoms, including 15 patients (57%) who experienced a complete recovery. Only two patients (11%) had severe persistent digestive symptoms.

#### 3.5.3. Dyspnea

Before TAE, 21 patients (81%) had dyspnea, including 16 with symptoms that caused problems in everyday life. A total of 14 of these patients (66%) reported improvement after TAE.

## 4. Discussion

TAE using microspheres and coils is a safe and an effective treatment for patients with diffuse and symptomatic PLD. These results confirm the findings of previous studies that used different embolization techniques [14,26,27,28,29] (Table 5).

The rate of absence of clinical amelioration and cyst volume reduction was 27% in our cohort compared to 0% reported by Ubara et al. [13], 15% reported by Wang et al. [14], 20% reported by Takei et al. [26], 34% reported by Park et al. [15], 40% reported by Sakuhara et al. [16], and 69.6% reported by Yang et al. [29]. These differences could be due to differences in treatment methodologies, including revascularization of embolized arteries, the development of extrahepatic collaterals stimulated by the cyst and parenchymal ischemia, and the development of intrahepatic collaterals when only one lobe was embolized [16].

These possibilities prompted us to perform a second TAE if the first was ineffective. The second TAE involved meticulously searching for and embolizing extrahepatic collaterals to ensure that there was no residual flow in the embolized artery. Revascularization of the hepatic artery seems to be the principal cause of failure [14,28]. The combination of distal embolization using microspheres and proximal occlusion of the segmental artery using microcoils seems to be very effective for producing irreversible occlusion of the targeted artery. 

Selecting the appropriate embolic agent is important. Recanalization is favored by the presence of many intrahepatic collateral vessels (peribiliary vascular plexus) and extrahepatic collateral vessels (omental artery, gastric artery, inferior phrenic artery, etc.) [31].

Different embolic materials have been used in previous studies, with coils being the most common [28]. Liquid adhesives and glue have also been used for distal and proximal embolization with good efficacy [14,30], however, there are risks for polymerization [32] and reflux in non-target areas [14,30], which can cause complications, such as biloma [14]. Calibrated microspheres are more precise and suitable than glue because of their ease of delivery from the inserted microcatheter and the low levels of associated inflammation compared to other liquid embolic agents [33,34].

In this study, the reduction in hepatic volume was −12.6% 3 months after embolization and −27.8% at the last follow-up more than 2 years after TAE are similar to those reported in the literature, that is, −7% to −21% with coils after 1 year [13,26], −25.7% and −29.3% for glue associated with lipiodol [30], and −15% after 1 year for polyvinyl alcohol particles and coils [15].

Neijenhuis et al. [35] showed that symptom reduction is a better outcome parameter than cyst volume reduction for treatment success in patients treated by aspiration sclerotherapy. Indeed, cyst diameter reduction does not reflect treatment success in aspiration sclerotherapy from patients’ perspectives, while symptoms measured with the PLD-Q can be used as a reliable outcome measure.

This procedure shows a low rate of complications (12.5%) compared to surgical approaches (50%) [8,36] and good tolerance as soon as postembolization syndrome could be controlled. Cyst infection is a serious but rare complication that occurred in only 4% of our cases and has previously been reported at rates between 0% and 2% [37]. The mean duration of hospitalization for TAE is shorter than that required for hepatic surgery (4.25 days vs. 15 days, respectively) [8,11].

Surgery can be performed after one or two TAE procedures. Indeed, embolization did not complicate surgery, and the reduction in volume permitted better mobilization of the liver and resulted in less risk for arterial bleeding during surgery.

In our study, two patients (7%) underwent partial hepatectomy after TAE because of insufficient clinical results with good efficacy. Partial hepatectomy and cyst fenestration substantially improves symptom burden and quality of life in highly symptomatic polycystic liver disease patients [38] with 11.1% of major complication.

This study is not without limitations. This was a retrospective, single center study with a small population. Additionally, there was an absence of systematic late determination of liver volume, an absence of a surgical control group, and the use of two types of microspheres. Moreover, questionnaire PLD-Q was not used in this study, which was validated by Neijenhuis et al. [39]. Due to the retrospective nature of this investigation, there are missing data.

In conclusion, TAE appears to be a safe and effective noninvasive treatment for patients with symptomatic Gigot 2 and 3 PLD. This study demonstrates a progressive decrease in liver volume with clinical efficacy in 73% of patients after one or two TAE procedures. This approach can improve the standard of care for patients with symptomatic PLD and can be considered before more invasive surgical procedures, even in patients with renal insufficiency.

## Figures and Tables

**Figure 1 jpm-12-01624-f001:**
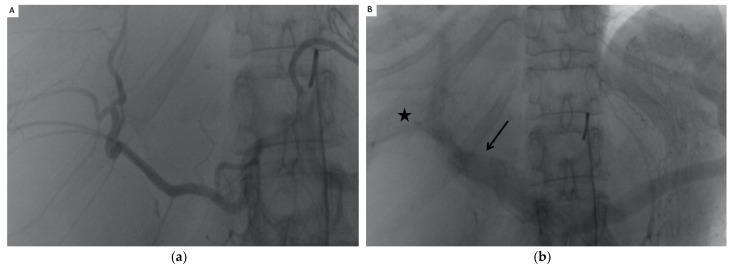
(**a**) arteriography of the coeliac artery showing well developed hepatic artery deviated by the cysts without parenchymography of the left liver. (**b**) Portography showing permeable splenic and portal vein, occlusion of the left portal and right anterior portal vein (arrow), and permeability of the right posterior portal vein (Star). (**c**,**d**): CT scan showing good correlation with the arteriography and portography. hepatic parenchyma is completely replaced by cysts in segment II, III and IV.

**Table 1 jpm-12-01624-t001:** Patient characteristics.

Patient Characteristics	Mean (Range) or N (%)
**Average age in years (range)**	52.3 (33–78)
<50 years	10 (39%)
>50 years	16 (61%)
**Gender**	
-Male	5 (19%)
Weight	86.6 (+/−12.8)
BMI	28.25 (+/−4.1)
-Female	21 (81%)
Weight	63,2 (+/−12.1)
BMI	23.4 (+/−2.4)
**Mean time follow-up (months)**	51 (6–98)
**Type of PLD**	
-associated with ADPK	20 (77%)
-PLD isolated	6 (23%)
**Laboratory**	
Creatinine	117.3 (+/−82)
>60 GFR	10 (38%)
<60 GFR.	13 (50%)
dialysis	3 (11%)
Urea	8.1 (+/−3.6)
AST	39.8 (+/−53)
ALT	33.2 (+/−36.5)
GGT	175.8 (+/−116.8)
Bilirubin	12.3 (+/−9.2)
Alcaline phosphatase	141.6 (+/−106)
TP	92.1 (+/−14)
Hemoglobin	12.7 (+/−1.5)
Platelets	230 (+/−77)

**Table 2 jpm-12-01624-t002:** Patients characteristics.

Patient Characteristics	Mean (Range) or N (%)
**Liver volume**	
Total liver volume	6436 cc (2965–13,470)
Right liver	4058 cc (2073–9566)
Left liver	2377 cc (848–5776)
**Symptoms**	
Abdominal Pain	24 (92%)
Dyspnea	21 (81%)
Dyspepsia	26 (100%)
**Previous treatment**	
Medical treatment	7 (27%)
Cyst sclerosis	11 (42%)
Fenestration or hepatectomy	0
**Anterior complications**	13 (50%)
Infection	12 (46%)
Hemorrhage	3 (11%)

**Table 5 jpm-12-01624-t005:** Summary of relevant studies on hepatic artery embolization in patients with polycystic liver disease.

Authors	Date	Patient	Embolic Material Used	Reduction in Liver Volume at 6 Months	Reduction in Liver Volume at 1 Year	Reduction in Liver Volume at 2 Years	Mean Liver Volume before Embolization (mL)	Clinical Success
Ubara et al. [13]	2004	1	Coils			−46%	12,364	100%
Takei et al. [27]	2007	30	Coils			−21.2%	7882	80%
Park et al. [15]	2009	3	PVA and Coils		−15%		9490	66%
Wang et al. [14]	2012	21	NBCA and lipiodol	No significative difference	−25.7%		8270	85.3%
Hoshino et al. [26]	2014	221	Coils	−5.3%	−9.2%		7058	-
Yang et al. [29]	2016	18	Coils	−7.6%			7767	31.4%
Zhang et al. [30]	2017	23	NBCA and lipiodol	−16.3%	−29.7%	−29.3%	8070	86%
Sakuhara et al. [16]	2019	5	Tris Acryl Gelatin microsphere	−5.5%	−6.7%		7406	60%

## Data Availability

The data presented in this study are available on request from the
corresponding author.

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
