# Peer review of "The Safety and Efficacy of Hepatic Transarterial Embolization Using Microspheres and Microcoils in Patients with Symptomatic Polycystic Liver Disease"

_jpm, 2022, doi:10.3390/jpm12101624_

Round 1
Reviewer 1 Report
THE WORK IS VERY INTERESTING BUT IT NEEDS SOME CORRECTIONS:
title highlights article’s purpose, that is safety and efficacy; the primary endpoint is the measurement of subjective wellbeing after TAE intervention. keeping these data, I would create a different exposure order, demonstrating the effectiveness of the treatment first (as described in the results) and only after the subjective well-being
IN INTRODUCTION: For the sake of completeness, I would point out that polycystic liver disease is part of a more varied cystic disease, which requires a precise diagnostic path(1).
IN MATERIALS AND METHODS: please briefly describe the questionnaire(2, 3)
IN RESULTS: I would better analyze the questionnaire: for example, I would assess the scale of pain, the feeling of well-being, highlighting in which area (pain, sense of wellness, etc) is more evident the benefit of this radiological treatment
IN DISCUSSION, I’d like you insert and comment, at the end, two recent papers:
1. Neijenhuis, M. K. Et al(4): authors analyzed quantitative assessment of symptom reduction is a better outcome parameter than cyst volume reduction for treatment success in patients treated by aspiration sclerotherapy and they concluded that: Cyst diameter reduction does not reflect treatment success in aspiration sclerotherapy from patients' perspective, while symptoms measured with the PLD-Q can be used as a reliable outcome measure
2. Bernts, L. H. P. et al(5): authors aimed to assess change in symptom relief and quality of life 6 months after partial hepatectomy and cyst fenestration in polycystic liver disease patients and they obtained that Partial hepatectomy and cyst fenestration substantially improves symptom burden and quality of life in highly symptomatic polycystic liver disease patients.
the citation of these two recent articles, (although they analyze different therapeutic techniques from the one used in this work), gives a more complete picture of the improvement of symptoms in polycystic liver disease
REFERENCES
1. Lantinga MA, Gevers TJ, Drenth JP. Evaluation of hepatic cystic lesions. World J Gastroenterol. 2013;19(23):3543-54.
2. Ware JE, Jr., Sherbourne CD. The MOS 36-item short-form health survey (SF-36). I. Conceptual framework and item selection. Medical care. 1992;30(6):473-83.
3. McHorney CA, Ware JE, Jr., Lu JF, Sherbourne CD. The MOS 36-item Short-Form Health Survey (SF-36): III. Tests of data quality, scaling assumptions, and reliability across diverse patient groups. Medical care. 1994;32(1):40-66.
4. Neijenhuis MK, Wijnands TFM, Kievit W, Ronot M, Gevers TJG, Drenth JPH. Symptom relief and not cyst reduction determines treatment success in aspiration sclerotherapy of hepatic cysts. Eur Radiol. 2019;29(6):3062-8.
5. Bernts LHP, Neijenhuis MK, Edwards ME, Sloan JA, Fischer J, Smoot RL, et al. Symptom relief and quality of life after combined partial hepatectomy and cyst fenestration in highly symptomatic polycystic liver disease. Surgery. 2020;168(1):25-32.
Author Response
THE WORK IS VERY INTERESTING BUT IT NEEDS SOME CORRECTIONS:
title highlights article’s purpose, that is safety and efficacy; the primary endpoint is the measurement of subjective wellbeing after TAE intervention. keeping these data, I would create a different exposure order, demonstrating the effectiveness of the treatment first (as described in the results) and only after the subjective well-being
IN INTRODUCTION: For the sake of completeness, I would point out that polycystic liver disease is part of a more varied cystic disease, which requires a precise diagnostic path(1).
Authors: We add on introduction:
Molecular genetic testing is available to look for mutations in the SEC63, LRP5, PRKCS, GANAB, ALG8, SEC61B PKD1, PKD2 and PKHD1.
IN MATERIALS AND METHODS: please briefly describe the questionnaire(2, 3)
Authors: We add:
The SF 36 questionnaire is a health survey frequently used on clinical studies to assess health related quality of life. It included 36 questions in eight domain scores of physical and mental function.
IN RESULTS: I would better analyze the questionnaire: for example, I would assess the scale of pain, the feeling of well-being, highlighting in which area (pain, sense of wellness, etc) is more evident the benefit of this radiological treatment
Authors: We write it accordingly.
IN DISCUSSION, I’d like you insert and comment, at the end, two recent papers:
- Neijenhuis, M. K. Et al(4): authors analyzed quantitative assessment of symptom reduction is a better outcome parameter than cyst volume reduction for treatment success in patients treated by aspiration sclerotherapy and they concluded that: Cyst diameter reduction does not reflect treatment success in aspiration sclerotherapy from patients' perspective, while symptoms measured with the PLD-Q can be used as a reliable outcome measure
- Bernts, L. H. P. et al(5): authors aimed to assess change in symptom relief and quality of life 6 months after partial hepatectomy and cyst fenestration in polycystic liver disease patients and they obtained that Partial hepatectomy and cyst fenestration substantially improves symptom burden and quality of life in highly symptomatic polycystic liver disease patients.
the citation of these two recent articles, (although they analyze different therapeutic techniques from the one used in this work), gives a more complete picture of the improvement of symptoms in polycystic liver disease
REFERENCES
- Lantinga MA, Gevers TJ, Drenth JP. Evaluation of hepatic cystic lesions. World J Gastroenterol. 2013;19(23):3543-54.
- Ware JE, Jr., Sherbourne CD. The MOS 36-item short-form health survey (SF-36). I. Conceptual framework and item selection. Medical care. 1992;30(6):473-83.
- McHorney CA, Ware JE, Jr., Lu JF, Sherbourne CD. The MOS 36-item Short-Form Health Survey (SF-36): III. Tests of data quality, scaling assumptions, and reliability across diverse patient groups. Medical care. 1994;32(1):40-66.
- Neijenhuis MK, Wijnands TFM, Kievit W, Ronot M, Gevers TJG, Drenth JPH. Symptom relief and not cyst reduction determines treatment success in aspiration sclerotherapy of hepatic cysts. Eur Radiol. 2019;29(6):3062-8.
- Bernts LHP, Neijenhuis MK, Edwards ME, Sloan JA, Fischer J, Smoot RL, et al. Symptom relief and quality of life after combined partial hepatectomy and cyst fenestration in highly symptomatic polycystic liver disease. Surgery. 2020;168(1):25-32.
Authors: We gratefully thanks you for this addition bibliography and we added this on the study.
Reviewer 2 Report
There are some minor comments.
It would be better to measure hepatic cystic volume and hepatic parenchyma volume, respectively, and compare them before and after TAE.
It would be better to add Legends of Figures 1 and 2.
Author Response
There are some minor comments.
It would be better to measure hepatic cystic volume and hepatic parenchyma volume, respectively, and compare them before and after TAE.
Authors: Cystic volume only is difficult to measure correctly.
It would be better to add Legends of Figures 1 and 2.
Authors: Sorry, here there are:
Figure 1. a arteriography of the coeliac artery showing well developed hepatic artery deviated by the cysts without parenchymography of the left liver. b Portography showing permeable splenic and portal vein, occlusion of the left portal and right anterior portal vein (arrow), and permeability of the right posterior portal vein.(Star) c and d: CT scan showing good correlation with the arteriography and portography. hepatic parenchyma is completely replaced by cysts in segment II, III and IV.
Figure 2 Computed Tomography (CT) from a patient with polycystic liver disease before and after transarterial embolization. A-b Pre-embolization CT showing a voluminous polycystic liver with a compression of the stomach by the cysts. C-d CT at 3 months showing coils and reduction of cysts volume. e-f: CT at 4 years showing a significant liver volume reduction of 29% with a decompression of the stomach.
Reviewer 3 Report
I am grateful to the editors of the JCM journal for the opportunity to get acquainted with this interesting study. Treatment of symptomatic polycystic liver disease (PLD) is a challenging problem of modern medicine. Pharmaceuticals (such as somatostatin receptor antagonists, mTOR Inhibitor and others) and traditional surgical procedures (such as puncture, fenestration, and hemihepatectomy) are not effective enough in the treatment of PLD. In addition, surgical interventions are accompanied by significant injury and may cause serious complications. In this regard, the use of mechanochemical transcatheter arterial embolization (TAE) seems to be an reasonable alternative to surgery. According to the literature, TAE is an effective treatment for PLD and is associated with a minimal complication rate. The work presented for the review proves this. Its authors objectively proved the clinical efficacy of TAE, demonstrated its safety, and suggested that recanalization of the target artery is the main cause of the treatment failure. In this regard, the authors recommend a combination of chemical and mechanical embolization to obtain the best outcome. An important conclusion is the authors' statement that TAE can be used as an intervention of choice before surgery.
Despite the obvious merits of the study, I have a few comments aimed at improving the quality of the manuscript and its perception by readers.
1. Materials and methods.
a) Please provide the inclusion/exclusion criteria. Only the indications for treatment are presented, while the inclusion/exclusion criteria must also be stated.
b) You specify follow-up periods of 3 months and more than 2 years. More than 2 years is a very vague time period. I propose to clarify, namely, to define the following time frames: 3 months, 2 years, 2-5 years, more than 5 years. Or specify only a timeframe of two years.
c) Please specify if the contrast enhancement was used while performing CT? What CT criteria were used to verify the diagnosis of PLD?
d) I did not find a description of the Figure 1. Please add a description of the figuure. What does the arrow point to? What are these arteries? Please indicate cysts and their size with arrows, etc.
e) Please indicate the diameter and length of the coils used. What was the average volume of microspheres injected (in milliliters)?
f) Please specify what laboratory tests were performed after TAE?
g) Please indicate whether you used any instrumental methods for assessing TAE (ultrasound, CT).
2. Results
a) In Table 1, please present the PLD symptoms (abdominal pain, distention, dyspepsia, and dyspnea) with the prevalence rate for each symptom. In the same table, please indicate the previous treatment of patients: how many patients received it, and what kind of treatment was carried out.
b) What extrahepatic arteries were embolized? Were these large vessels or tributaries?
c) Please specify the symptoms of post-embolization syndrome (PES) and the timing of its relief. Are there any options to prevent PES after TAE?
d) Please add a description to the Figure 2 and demonstrate positive changes due to the treatment performed. Please specify in the description the terms when CT was performed. You can also add a percentage decrease in the volume of the liver and in the size of the cysts.
e) Was there a positive trend in laboratory parameters?
f) Were there any peculiarities in the management of patients with renal insufficiency?
3. Please separate the Limitations section and state also there that the number of patients included in the study was small.
After corrective measures the manuscript can be accepted for publication.
Author Response
I am grateful to the editors of the JCM journal for the opportunity to get acquainted with this interesting study. Treatment of symptomatic polycystic liver disease (PLD) is a challenging problem of modern medicine. Pharmaceuticals (such as somatostatin receptor antagonists, mTOR Inhibitor and others) and traditional surgical procedures (such as puncture, fenestration, and hemihepatectomy) are not effective enough in the treatment of PLD. In addition, surgical interventions are accompanied by significant injury and may cause serious complications. In this regard, the use of mechanochemical transcatheter arterial embolization (TAE) seems to be an reasonable alternative to surgery. According to the literature, TAE is an effective treatment for PLD and is associated with a minimal complication rate. The work presented for the review proves this. Its authors objectively proved the clinical efficacy of TAE, demonstrated its safety, and suggested that recanalization of the target artery is the main cause of the treatment failure. In this regard, the authors recommend a combination of chemical and mechanical embolization to obtain the best outcome. An important conclusion is the authors' statement that TAE can be used as an intervention of choice before surgery.
Despite the obvious merits of the study, I have a few comments aimed at improving the quality of the manuscript and its perception by readers.
- Materials and methods.
- a) Please provide the inclusion/exclusion criteria. Only the indications for treatment are presented, while the inclusion/exclusion criteria must also be stated.
Authors: We write it accordingly. Exclusion criteria included liver cyst infection and the decision of hepatic transplantation taken before TAE.
- b) You specify follow-up periods of 3 months and more than 2 years. More than 2 years is a very vague time period. I propose to clarify, namely, to define the following time frames: 3 months, 2 years, 2-5 years, more than 5 years. Or specify only a timeframe of two years.
Authors: We write it accordingly.
- c) Please specify if the contrast enhancement was used while performing CT? What CT criteria were used to verify the diagnosis of PLD?
Authors: We write it accordingly. CT with or without contrast injection
- d) I did not find a description of the Figure 1. Please add a description of the figuure. What does the arrow point to? What are these arteries? Please indicate cysts and their size with arrows, etc.
Authors: Sorry it was a mistake.
Figure 1. a arteriography of the coeliac artery showing well developed hepatic artery deviated by the cysts without parenchymography of the left liver. b Portography showing permeable splenic and portal vein, occlusion of the left portal and right anterior portal vein (arrow), and permeability of the right posterior portal vein.(Star) c and d: CT scan showing good correlation with the arteriography and portography. hepatic parenchyma is completely replaced by cysts in segment II, III and IV.
- e) Please indicate the diameter and length of the coils used. What was the average volume of microspheres injected (in milliliters)?
Authors: We don’t have these data.
We add this sentence: Diameter of coils were between 2 and 6mm.
- f) Please specify what laboratory tests were performed after TAE?
Authors: Laboratory data included Creat, Urea, AST, ALT,GGT, Bilirubin, and Alcaline phosphatase.
- g) Please indicate whether you used any instrumental methods for assessing TAE (ultrasound, CT).
Authors: into the right or left femoral artery under ultrasound guidance,
We didn’t use CT to perform TAE.
- Results
- a) In Table 1, please present the PLD symptoms (abdominal pain, distention, dyspepsia, and dyspnea) with the prevalence rate for each symptom. In the same table, please indicate the previous treatment of patients: how many patients received it, and what kind of treatment was carried out.
Authors: I don’t understand, theses data are summarized on table 1, do you want to mix tables 1 and 2 ?
- b) What extrahepatic arteries were embolized? Were these large vessels or tributaries?
Authors: we add
Extra-hepatic arteries embolized were three left gastric arteries.
- c) Please specify the symptoms of post-embolization syndrome (PES) and the timing of its relief. Are there any options to prevent PES after TAE?
Authors: We add:
PES is characterized by moderate to severe epigastric pain, fever, severe nausea and vomiting that appear early after embolization. Biologically, there is a biological inflammatory syndrome associated with elevated transaminases and bilirubin. All these abnormalities are transient and spontaneously resolved in a few days.
- d) Please add a description to the Figure 2 and demonstrate positive changes due to the treatment performed. Please specify in the description the terms when CT was performed. You can also add a percentage decrease in the volume of the liver and in the size of the cysts.
Authors: Sorry it was a mistake.
Figure 2 Computed Tomography (CT) from a patient with polycystic liver disease before and after transarterial embolization. A-b Pre-embolization CT showing a voluminous polycystic liver with a compression of the stomach by the cysts. C-d CT at 3 months showing coils and reduction of cysts volume. e-f: CT at 4 years showing a significant liver volume reduction of 29% with a decompression of the stomach.
- e) Was there a positive trend in laboratory parameters?
Authors: there is a transitory biological inflammatory syndrome associated with elevated transaminases and bilirubin.
- f) Were there any peculiarities in the management of patients with renal insufficiency?
Authors: None
- Please separate the Limitations section and state also there that the number of patients included in the study was small.
Authors: We write it accordingly.
The main limitations of this study are its retrospective nature and the small population. Also, the absence of systematic late determination of liver volume, the absence of a surgical control group, and the use of two types of microspheres.
After corrective measures the manuscript can be accepted for publication.
Round 2
Reviewer 1 Report
THANK YOU FOR YOUR CORRECTIONS
Author Response
thank you for your very interesting work. You have responded well to the remarks
of the reviewers. I ask you to make a last small effort before publication to make
your article more readable:
- in the result section, review the order:
3.1: population, 3.2 technical success, 3.3 safety, 3.4 reduction hepatic volume, 3.5
clinical efficacy
Authors : We write it accordingly
- in the "reduction hepatic volume" section, add the time frame for the reduction in
volume, if I have understood correctly 3 months. If not, please clarify when the
reduction in volume took place on average.
Authors : We write it accordingly
The mean decrease in total liver volume at 3 months was –12.6% (±â€…8.01%) compared to the pre-TAE value, or a mean loss of –855 cc (p < 0.01). CT was performed more than 2 years after TAE in 12 patients, and the mean decrease in total liver volume ratio was –27.8% (p < 0.01) (Table 4).